# Advances in Cardiac Tissue Engineering

**DOI:** 10.3390/bioengineering9110696

**Published:** 2022-11-16

**Authors:** Takahiro Kitsuka, Fuga Takahashi, James Reinhardt, Tatsuya Watanabe, Anudari Ulziibayar, Asigul Yimit, John Kelly, Toshiharu Shinoka

**Affiliations:** 1Center for Regenerative Medicine, Nationwide Children’s Hospital, Columbus, OH 43205, USA; 2Department of Cardiothoracic Surgery, Nationwide Children’s Hospital, Columbus, OH 43205, USA; 3Department of Surgery, The Ohio State University Wexner Medical Center, Columbus, OH 43210, USA

**Keywords:** tissue engineering, iPS-derived cardiac myocyte, biodegradable scaffolds, synthetic polymers, 3D culture, vascularization, 2D culture, 3D printing, clinical trials

## Abstract

Tissue engineering has paved the way for the development of artificial human cardiac muscle patches (hCMPs) and cardiac tissue analogs, especially for treating Myocardial infarction (MI), often by increasing its regenerative abilities. Low engraftment rates, insufficient clinical application scalability, and the creation of a functional vascular system remain obstacles to hCMP implementation in clinical settings. This paper will address some of these challenges, present a broad variety of heart cell types and sources that can be applied to hCMP biomanufacturing, and describe some new innovative methods for engineering such treatments. It is also important to note the injection/transplantation of cells in cardiac tissue engineering.

## 1. Introduction

Heart failure is one of the leading causes of death worldwide, especially due to the increasing number of patients with end-stage heart failure, despite the many treatments available MI, in particular, is a disease with high global mortality and morbidity caused by the permanent loss of myocardial cells (MC) leading to progressive pathological left ventricular (LV) remodeling and heart failure. Heart muscle can not regenerate; instead, the regenerative capacity of the adult heart declines rapidly after birth, with less than 1% of the heart’s MCs being replaced each year [1]. Thus, heart transplantation becomes the only possible form of treatment with drastic improvement, for pharmacological treatments are not a long-term solution to heart failure and do not regenerate the myocardium [2]. However, heart transplantation, considered to be the standard treatment for end-stage heart failure, is limited by the serious shortage of donors and the need for immunosuppression [3,4]. Even after surgery, systemic circulation may be insufficient for patients with severe MI, making the ventricular assist device treatment the only treatment left if conditions worsen. The urgent need for alternative strategies to limit post-injury cardiac remodeling has led to many clinical studies on cell therapy. 

A particular study developed cell myocardoplasty; a cell administration method where one injects pluripotent cells directly into the heart muscle instead of transplanting a solution directly to the coronary artery by using a syringe [5]. Injecting embryonic stem cells (ESCs); induced pluripotent stem cells (iPSCs); and other cell sources have successfully demonstrated an improvement in the function of the damaged heart; and subsequently some regeneration of the new myocardium [6]. However; cell myocardoplasty has a few downsides that plague its potential. Study results show that a high percentage of cells did not differentiate into the cardiomyocytes (CM) as expected; and were not retained at the site of injury. Moreover; direct injection of stem cells into the ischemic heart does not improve cardiac function because the microenvironment of the ischemic heart does not adequately support the survival of the transplanted cells. It has been reported that more than 70% of cells die in the first 48 h after direct injection. The remaining 30% of the surviving cells gradually die within the next few days due to hypoxia; inflammation; and/or fibrotic microenvironments [5]. Only a few studies with this method; such as one by Liu. et al. that used human embryonic stem cell-derived cardiomyocytes to successfully recover the myocardium by ~10.6% have been published; and even then requires larger testing groups to be considered for clinical application [6]. A tissue engineering approach was developed to address the limitations of cell myocardoplasty. Tissue engineering combines the principles of engineering and life sciences to better understand the relationship between the structure and function of normal and pathological tissues; and allows for the production of biological tissues for drug testing; disease modeling; and further research for regenerative medicine. Currently; there are various approaches to cardiac tissue engineering; such as an injectable in situ delivery of cells; and in vitro engineering of contractile tissue constructs for transplantation [5]. Another approach includes manipulated patch constructs; which are often associated with higher engraftment rates and appear to support damaged myocardium more effectively than transplanted cells. In all; this field is moving forward with the research and development of artificial hCMP. 

## 2. Cell Types and Sources for Artificial hCMP Fabrication

Originally, many investigations in myocardial regenerative medicine have focused on CMs, cells difficult to proliferate and recover once damaged. CMs are responsible for generating a contractile force in the myocardium [7]. In the heart, the newborn ratio for the number and volume ratio of CMs to non-CMs is 7:3. As the heart grows to adult size. However, the volume ratio does not change, the ratio for the number of CMs to non-CMs becomes 3:7 with the non-CMs increasing in proportion [8]. The cause of this ratio change has been hypothesized to be the proliferation of non-CMs; from this, it is thought that the function of the heart is due to not only the CMs but also the interaction of non-CMs and CMs.

CMs, endothelial cells (ECs), smooth muscle cells (SMCs), and cardiac fibroblasts (CFs) have been previously used for hCMP production, in order to comprehensively reproduce the physical structures and signaling pathways present in native heart tissue. ESCs and iPSCs are the most readily available sources of CM for human strains because they can multiply indefinitely and differentiate into cells of different lineages. Both ESC-derived and iPSC-derived CM are also more structurally similar to neonatal cells than adult CM, hindering applications of hCMPs composed of immature iPSC-derived CM. However, clusters of other cell types (e.g., progenitor cells and spheroids) have also been incorporated into the heart patch and evaluated in preclinical models of myocardial injury [9]. It is also important to note how Zhao et al. is working on differentiating muscle cells into specific ventricular and atrial cells [9]. 

### 2.1. Skeletal Myoblasts (SMs)

SMs is the first discovered cell source for cell sheets and is now attracting many researchers for the treatment of acute myocardial infarction (AMI). In basic research, SM has been applied following MI in various animal models such as rats, hamsters, dogs, pigs, etc. [10]. SMs have an advantage as they can be autologously sourced, thereby reducing concerns of eliciting an immune response and they possess ischemic resistance, the ability to differentiate into non-myocyte lines, and high proliferative potential. However, myoblasts do not express myocardium-specific contractile proteins or connexin 43 when transplanted into the heart, causing them to be electrically isolated and beat asynchronously with the recipient’s heart. SMs cannot form a gap junction with CMs, which can cause arrhythmias. Myoblasts are generally known to compensate for damaged skeletal muscle by proliferating and differentiating. When myoblasts are implanted as part of cardiac tissue engineering strategies, part of their therapeutic benefit may be due to paracrine signaling due to their production and release of growth factors and cytokines. Regarding the release of growth factors and cytokines, a study utilizing myoblast sheets implanted in the heart tissue in a chronic MI rat model hypothesized that various cytokines such as hepatocyte growth factor (HGF), vascular endothelial growth factor (VEGF), and stromal-derived factor-1 (SDF-1) were released from transplanted cells [11]. This hypothesis was largely due to the high expression of insulin growth factor (IGF), improved cardiac function, and the increased expression of HGF and VEGF in the recipient myocardium [12]. Furthermore, a high concentration of SDF-1, a chemokine for bone marrow-derived cells, as well as integration of some stem cell-derived factors such as c-Kit and Sca-1 positive cells were reported. Uchiha et al. reported that laminin α 2 secretory fibroblasts enhance the therapeutic effect of skeletal myoblast sheets by inhibiting the detachment of transplanted myoblasts from transplanted myocardium [13] (Table 1). For this, it was suggested that myoblast sheet transplantation may improve cardiac function self-repair by inducing stem cells with growth factors and chemokines. On another note, previous non-clinical studies have shown that transplanted myoblasts cannot be detected histologically six months after transplantation. When transplanted myoblast sheets fall off in the late stages after transplantation, their function is maintained and the myoblasts express the hypoxia-inducible factor-1 (HIF-1) gene at a high rate [14]. HIF-1 is responsible for angiogenesis at the transplant site as well as the induction and recruitment of bone marrow mesenchymal stem cells (MSCs) [15,16]. Although clinical trials utilizing cell sheets of skeletal myoblast have demonstrated safety and feasibility before clinical use one should consider improving the current SM-derived cell sheet by combining it with synchronous contraction devices to avoid the unknown effect of myoblast sheets on advanced heart failure and the potential for fibrosis.

### 2.2. MSCs

MSCs’ immunomodulatory abilities and growth factor secretion capabilities show promise in the field of tissue engineering, leading to many clinical trials for their use in the extensive treatment of ischemic heart disease [17]. A few studies have already suspended bone marrow MSCs on 3D hydrogels to test MI in rat models. After 8 weeks, the autologous MSCs significantly improved LV function, promoted angiogenesis at the periphery of the infarction, and reduced infarction volume as well as suppressed apoptosis of host cardiomyocytes 4 weeks after transplantation [18]. Transplanted MSC sheets using cells obtained from bone tissue, adipose tissue, and menstrual blood also showed some myocardial formation and dramatic paracrine effects that contribute to angiogenesis, cardioprotection, improved LV function, and myocardial repair (Figure 1). MSCs can be applied not only in heart sheets but also in regenerative blood vessels, for the cells contribute to the relaxation of the immune response of tissue-engineered vascular grafts (TEVGs) [19]. These applications confirm the safety of MSC and its partial mechanism of action.

### 2.3. Cardiomyocytes (CMs) (Fetal Myocardium and iPSCs)

The myocardium is similar to a skeletal muscle but is instead made up of many CMs, where each CM is electrically connected to adjacent CMs by gap junctions allowing for synchronization of contraction and relaxation. CMs, like skeletal muscle, are difficult to differentiate and mass-culture because CMs in healthy adults are rarely available. Several research teams have reported that cell sheets derived from fetal or neonatal CMs can be used to treat MI in animals. Pioneering studies, using fetal CMs from 15-day-old mouse embryos have demonstrated that donor cells survive after transplantation and administration to the heart [7]. While this approach holds promise, the use of fetal or neonatal CMs in humans would present a complex ethical dilemma.

In 1981, the discovery of pluripotent stem cells (PSCs) using mouse ESCs led to the creation of myocardium using PSCs with the characteristics of infinite proliferation and highly efficient CM differentiation [20]. ESCs and PSCs are both created by extracting the internal cell mass of an early embryo, and the products have the versatility to differentiate into all cells in the living body; a quality very advantageous as cell sources for the cell sheets [21]. A cell sheet using ESC-derived CMs was developed, where a new cardiovascular cell differentiation system can be established from mouse ESCs, to collect CMs and vascular cells in vitro [22]. However, the engraftment efficiency of the ESC-derived (CTS) was considerably lower 4 weeks after transplantation, requiring further studies to investigate another cell source for better cell survival and myocardial regeneration. Compared to the ESCs, iPSCs that Yamanaka et al. developed can be generated from a patient’s somatic cells without ethical issues, and therefore have been studied for the treatment of MI [23].

It is also worth noting how electrical stimulation or excitation-contraction coupling can affect the amount and duration of action potentials in CMs, thus increasing the number of synchronous contractions. Ruan-Circ et al. conducted a study with collagen-based, bioengineered tissue created with CMs derived from hiPSCs; electrical pacing combined with static stress conditioning led to a ~1.34 mN/mm^2^ increase in force production, suggesting maturation [24]. Electrical stimulation can trigger several transcription factors such as NRF-1, GATA4, NFAT3, and cytochrome c, leading to increased growth and maturation in CMs [25]. However, electrical stimulation could lead to arrhythmias due to the changes in electrical coupling [26]. Moreover, most biomaterials used in 3D constructs are still not synchronized, requiring new materials such as graphene to be added [27,28,29].

### 2.4. Supporting Cells: Vascular Endothelial Cells, Fibroblasts, and SMCs

Blood vessels have a major role in the heart, transporting nutrients and oxygen using the flow of blood. Blood flow is essential in regenerative medicine due to the large size of myocardial tissue, which requires a functional vascular network within. Stevens et al. and others have shown that a cohesive heart patch composed of CMs, ECs, and fibroblasts derived from human ESCs develops vascular structures that are essential for successful transplantation in vivo. Arai et al. also used a combination of iCell, ECs, and fibroblasts to form a cardiac spheroid of interest. Based on this idea, Arai et al. hope that blood flow can be ensured by mixing human umbilical vascular ECs when creating structures based on iPSC-derived CMs [30]. Recently, cardiac fibroblasts have been successfully distinguished from human pluripotent stem cells(hPSCs); cells similar to natural cardiac fibroblasts in morphology, gene expression, and proliferation. hPSC-derived cardiac fibroblast cocultured with hPSC-CM would increase the rate of the action potential propagation compared to coculture with skin fibroblasts [29,31]. Optimal combinations/ratios of cell types for reproducing complex 3D environments of native heart tissue continue to be an active area of research. There is no doubt that both fibroblasts play an important role in scaffolds to create a large structure such as that of myocardial tissue. SMCs secrete several angiogenic factors, including basic fibroblast growth factors (bFGF), VEGFs, and HGFs [32], and a paracrine angiogenic effect is expected. SMC cells can also induce differentiation from iPSC, and various methods have been reported [33].

**Table 1 bioengineering-09-00696-t001:** Exploring several cell sources for the creation of hCMP, such as SMs, MSCs, Bone marrow cells, adipocytes, and CMs.

Cell Sources	Comments	Ref.
Skeletal myoblasts (SMs)	Source of progenitor cells, to repair in the event of MI. Activated in response to muscle damage, then expresses Myf-5 or MyoD, myogenin, and MRF4. Ability to expand in vitro, resist ischemia, and have myogenic differentiation. Surrounds the sarcolemma. Advantages include a reduced likelihood of an immune response reaction, resistance to hypoxic conditions, production of angionenic factors, and a contractle phenotype. On the other hand, arrhythmias could occur, whilst having a low survival rate and a high chance of rejection.	[4,34,35]
Mesenchymal stem cells (MSCs)	Located in the blood vessel wall, and difficult to distinguish due to lack of unique markers. Ability to release anti-apoptopic and pro-angiogenic factors, as well as inflammatory agents, in order to inhibit inflammatory reactions. Although MSCs have advantages including an immunosuppressive potential and easy harvesting, the lack of evidence for safety, as well as its profibrogenic potential holds these cells back.	[36,37,38,39]
Cardiomyocytes	Derived from some sources, although rarely available: neonatal animals, Sca-1 (+) and C-kit (+) cardiac progenitor stem cells (CPCs) from adult murine hearts, ESC/iPSC-derived pure cardiomyocytes. Release extracellular vesicles for regenerative ability, contributing to cell contraction and relaxation. Although cardiomyocytes do show promise, they are hard to culture ethically, so similar alternatives have to be considered.	[6,40,41]
Bone marrow cells	Ability to produce bioactive molecules while interacting with the immune system. Overall, increase the ability of compromised tissue to regenerate. Can also be used to create disease models. Advantages include an immunopriveleged profile, paracrine/proangiogenic effects, and reliability, for it has been tested in several clinical studies.	[42,43]
Adipocytes, adipose-derived stem cells	Different characteristics and density arise if harvested from different areas and cells. Several reports have been made on adipose-derived stem cells’ ability to differentiate into several lineages; endodermal, ectodermal, and mesodermal. Can secrete multiple growth factors and cytokines, for regenerative capabilities. Moreover, it is easy to obtain large numbers of them, by using liposuction.	[42,44,45]
Supporting cells: vascular endothelial cells, fibroblasts, and SMCs	Fibroblasts with the optimal combination of cell types and ratios will produce an improved scaffold; moreover, SMCs can secrete various factors and induce differentiation from IPSC.	

## 3. How to Create Sheets

Advances in stem cell biology have made it possible to use various cells such as adult stem cells and iPSCs, and advances in bio-3D printing have made it possible to create large tissue structures for transplantation. An approach utilizing scaffolds, a method of seeding a vesicle into a three-dimensional and porous scaffold using a biodegradable polymer material involves mixing and pouring cells and gel-like scaffolds (e.g., collagen) into a mold, then arranging the cells using a 3D printer and a small amount of gel. Another method, bioprinting, has been pursued as well where fiber-like cell-containing gels are bundled together. Spheroids and organoids-clumps of cells have also attracted attention. By seeding these cell masses in metal needles or metal devices, organizing then extracting them, one can create a three-dimensional structure that does not contain scaffolds (Figure 2a).

### 3.1. Cell Sheet Approach to Producing hCMP

Most clinical studies use sheets for their approach; for example, the cell sheet method developed by Professor Okano of Tokyo Women’s Medical University consists of synthesizing myocardial tissues with a pulsating ability, without utilizing cell supports [46]. The cell sheets are created using a special culture dish coated with poly N-isopropyl acrylamide. To create sheets with sheet-like cells, one must generate and transplant tissues by stacking each monolayer. Furthermore, a culture dish, with the ability to control cell desorption by changing the hydrophilicity according to surface temperature, would be used for the recovery of the cell sheet. At a normal culture temperature of 37 °C., it becomes a hydrophobic surface, and in this state, cell adhesion factors such as fibronectin adhere to the surface of the normal culture dish. When the temperature drop treatment (32 °C or lower) is performed, the surface of the instrument becomes hydrophilic, and the cells desorb from the surface of the culture dish together with membrane proteins. 

Because the cell sheet lacks exogenous/synthetic scaffolding, there are no concerns about the potential immunogenicity of the scaffold material. A major advantage of creating tissues this way is that they maintain adhesion proteins on the surface of the cell tissue body, so when transplanted into living organs, they have an admirable integration function with the transplanted organ. Furthermore, cells in adjacent layers can form connections between the layers that facilitate communication, including gap connections necessary for electrical bonding.

### 3.2. 3D Printing; Spheroids, Contractile Forces, and Tubular EHTs

Early research used tissue printing techniques, where the printed constructs worked well with the cardiac system and demonstrated high survival rates after 7 days of culture [47]. However, these scaffolding-based tissue engineering approaches also have some drawbacks; collagen and other biological molecules deteriorate the scaffolds rapidly, showing that further research is needed for appropriate tissue engineering and/or cell delivery techniques. 3D printing stacks up the previously mentioned spheroids, a method that leads to a higher function expression compared to two-dimensional cultures. Spheroids also promote the differentiation of the myocardium, promoting maturation [48]. Some of these spheroids use CMs, due to the advancement in IPS CM purification methods that can create cells with 99% purity [49]. In one study, transplanted spheroids composed of CPCs, expressing ISL1-LIM-homeodomain transcription factors differentiated into CM and EC, contribute to the formation of new blood vessels in the heart of infarcted mice [1]. On a new note, bioprinting techniques are gaining traction because they use CAD modeling to create large organs with controllable, native-like substances such as living cells and other biological materials [50]. Bioprinted hCMP without scaffolding is produced by loading spheroids one by one into an array of needles, fusing them, then removing the hCMP and culturing it until the holes in the needle are filled with the surrounding tissue. Here, we will continue to introduce structures with an iPS myocardium that does not use a scaffold.

Breckwoldt et al. developed hCMPs by differentiating human ESCs into hiPSC-CMs, a design that allows one to analyze human heart diseases by checking the EHT force strength [51].

A system for evaluating the 3D contractile force of the heart’s structure is important for new drug development. Although some research groups have already reported scaffold-based heart constructs, these scaffolding-based heart structures cannot accurately predict the heart’s drug response in vivo due to the interaction between the drug and scaffolding material [51]. Heart structures without scaffolding, such as patches and spheroids, have been manufactured but have not been reported to accurately assess the contractile force of the 3D heart structure. Although several studies have reported that contractile forces could be measured with the use of myocardial structures such as organoids, the contraction evaluation of said studies cannot be accepted because the shrinkage of the structure is very small and the reproducibility is poor [52]. Therefore, after creating a scaffolded three-dimensional (3D) tubular heart structure with a bio-3D printer, Arai et al. established an analysis system that can measure changes in needle tip movement as an index of the contractile forces.

Another study without the use of a scaffold consists of a tubular EHT (T-EHT), created using bio-3D printers with hiPSC-CO and needle arrays, which produced beating conduits for patients suffering from monoventricular disease. In this study, we created a hiPSC-derived cardiac organoid composed of hiPSC-derived CMs, human cord vein endothelial cells, and human fibroblasts [53]. A bio-3D-printed T-EHT has been implanted around the abdominal aorta as well as the inferior vena cava (IVC) of NOG mice. (Figure 2b) Muscle stripes were observed with MLC2a, MLC2v, and α-actinin staining of tissue, all of which are indicators of maturation of myocardial tissue. After the T-EHT, which had been in vivo for 4 weeks, was removed and cultured, we observed it to have a spontaneous beat. Although the cultured T-EHT demonstrated an increased beating rate when we applied a bipolar electrical pulse, abundant evidence had not been provided to prove whether the T-EHT supported blood flow. Although a few of such non-scaffold myocardial structures have been reported, further advances in research are awaited [54].

**Figure 2 bioengineering-09-00696-f002:**
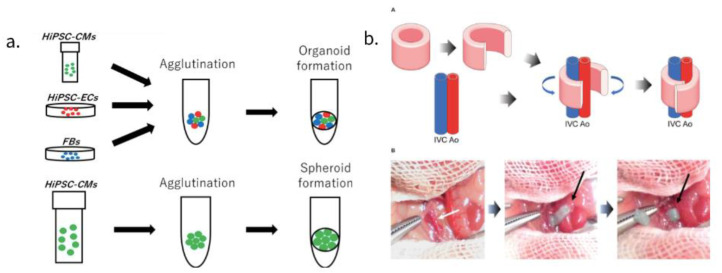
(**a**) A combination of HiPSC-CMs, HiPSC-ECs and fibroblasts can be utilized to create cardiac spheroids [2]. (**b**) Figure of transplantation procedure of T-EHT, as well as the transplantation procedure, where the aorta and IVC of mice are exposed [54].

## 4. Common Problems for All

### 4.1. Increased Thickness of hCMP

Few studies have been conducted using appropriately sized constructs, despite significant improvements in protocols used in hCMP manufacturing. After transplantation, the diffusion of oxygen and nutrients from the vascular system limits the thickness of the hCMP. This diffusion can be controlled by mixing molecular crystals (e.g., sucrose) in the matrix solution and leaching them after the matrix has solidified [55]. However, a dense internal vascular network that can bind to the natural circulation is also necessary for the creation of an HCMP with clinically appropriate thickness. Vascularization can be increased with the combination of blood vessels and other cell types (EC, SMC, and/or fibroblasts) during production and/or through nanoparticles. However, more research is required, for even patches with a relatively large surface area (for example, 8 cm^2^) are still made relatively thin (1.25 mm), jeopardizing clinical application because they cannot mimic native myocardial characteristics such as action potentials [56].

In theory, layer-by-layer assembly technology would allow one to produce hCMP of any thickness by simply stacking the required number of individual CM sheets. Incorporating graphene oxide(GO)-based thin films would not only enhance the formation of physical and electromechanical connections between layers, but can also improve electrical bonding, maturation, and cell tissue growth. Moreover, by culturing heart cells on a 2D cell mesh that incorporates a micromachined hook-and-loop system, the hooks and loops in adjacent layers would have the ability to work together for the hCMP to contract in response to electrical stimuli [57].

#### Designed Vascular Network

Rather than relying solely on infiltration from the original circulatory system, thicker hCMP may require some artificial blood vessels before transplantation. Angiogenesis can be induced by encapsulating a sacrificial gelatin mesh in a scaffolding material and then melting the gelatin mesh to leave a network of interconnected microfluidic channels. When seeded with human microvascular EC, the sacrificial scaffolding generates a rudimentary endothelial network [58]. Another strategy for artificial blood vessels involves the use of sustained-release formulations of the angiogenic factor thymosine β; by promoting and inducing the growth of blood vessels from exotransplanted arteries and veins, as well as forming capillary beds within hydrogel scaffolds, endogenous angiogenesis processes can be mimicked. Vascular growth can also be directed using micropatterned polyglycerin sebacic acid scaffolding, where after transplantation, host blood cells infiltrate microvessels as the scaffold degrades. Micropatterning has also been used to organize ECs into “codes’’ to induce the formation of capillaries that integrate with host tissue.

### 4.2. hCMP Constructs

#### 4.2.1. hCMP Delivery Method

Cell and tissue delivery methods can be classified into invasive and non-invasive approaches, both of which are often characterized by low engraftment rates [59]. Implantation of hCMP is primarily an invasive delivery, requiring open-heart surgery and suturing/attaching the patch to the epicardium. Due to the need for dedicated facilities and highly trained staff to perform open-heart surgery, the invasive delivery method is not optimal for repeated applications. Intramyocardial cell injections into the damaged myocardium, on the other hand, have shown relatively lower cell retention. Studies have shown that repeated cell injections increase therapeutic efficacy, as a result, invasive delivery of cells and hCMP has been a focus for the clinical translation of these therapies [60]. Recently, an epicardial delivery of hydrogels through pericardial cavities was conducted; a pericardial device (catheter) delivered hydrogel components through separate (coaxial) lumens and combined them to form a stable hydrogel structure between the pericardial and epicardial layers [61]. This method helps to minimize the risk of extensive myocardial injury, thrombotic blockage, and arrhythmias. Injectable hydrogels are also being tested as a minimally invasive (or non-invasive) delivery approach for heart patch systems. In a study where cell-free alginate-based hydrogels were injected into MI patients to demonstrate maintenance of LV index and ejection fraction, the hydrogels in patients with advanced HF lead to approximately 9% of deaths within 30 days of injection [62]. Because the control group reported 0 deaths, one can conclude that although significant progress has been made in the development of injectable heart patch systems, more effort is needed for their further improvement in efficient clinical use. These include further enhancement of biochemical properties in hydrogel constructs to mimic natural tissue, improvement of cell viability and biomolecular activity, a more controlled degradation, stronger immune response, and enhanced in vivo tracking of the patches.

Alternatively, non-invasive intravascular (i.e., intravenous) delivery shows recirculation and redistribution of injected cells into other organs other than the target site, which, as previously discussed, leads to improved cardiac function due to paracrine signaling mechanisms. This method has the advantage of being non-invasive and has the potential for repeated administration, therefore being considered the appropriate selection from a clinical perspective.

#### 4.2.2. Animal Models for Testing hCMP

Preclinical studies have utilized a variety of animal models, including mice, rats, guinea pigs, pigs, and non-human primates (Table 1). The selection of appropriate animal models with a high safety profile and validity of treatment outcomes is critical to the purposes of clinical translation. Although initial studies for the therapeutic effect of transplanted human cells and heart tissue were conducted on immunodeficient rodents, including athymic rats and severely impaired mice, disagreements in anatomy and physiology between rodents and humans weaken the reliability of treatment results. 

Subsequent preclinical studies have utilized large animal models such as non-human primates and pigs to demonstrate remuscularization and therapeutic effects of concern for ventricular arrhythmias. Large animal models are more reliable than compared to rodent models, although the high costs associated with large animal research will limit their use [63,64].

#### 4.2.3. Addressing Obstacles of hCMP

The ultimate goal of a heart patch transplant is to replace the damaged heart muscle with an extrinsically functioning myocardium. Preclinical studies from mice, rats, guinea pigs, pigs, and non-human primates showed some myocardial remuscularization of fibrous scar tissue. However, the studies faced several challenges, with one being the large amounts of exogenous CM (and/or other heart cells) required to replenish lost tissue. Another challenge is ensuring long-term graft retention for maximum therapeutic effect. Genetic engineering can be utilized to increase cell retention, where overexpression of CCND2 (cyclin D2), a cell cycle activator, increases the cell cycle activity and growth rate of hiPSC-CM, thus improving engraftment rates from an average of 10% to 25%. The lack of conductivity between the hCMP and the host tissue proves to be another challenge. One study utilized super-aligned carbon nanotubes during the manufacture of cardiac tissue, enhancing electrophysiological uniformity due to the anisotropic conductivity of their aligned structures. Another study demonstrated that electrospinning nanofiber scaffolds with enhanced conductivity promoted the electrical bonding of patches, indicating the potential for the manufacture of clinically relevant hCMP products. As discussed in the previous section, one major aspect that needs to be carefully studied when addressing the electromechanical integration of exogenous CM and host myocardium is the high probability of causing arrhythmias (Figure 3). 

### 4.3. Safety Concerns

The occurrence of arrhythmias is a serious safety concern for the clinical applicability of heart patch therapy. Large animal models are being used to study arrhythmias in cardiac patch studies, due to their sinusoidal waves providing a more accurate representation of the physiology, function, and anatomical structure of the human heart. Although preclinical studies of non-human primates showed transient non-fatal arrhythmias 2 weeks after transplantation, after which these arrhythmias decreased, ventricular arrhythmias in pig models were more frequent and fatal (2 out of 7 pigs) [65]. It is currently believed that impaired impulses at the interface between the transplanted CM and host myocardium may induce ventricular arrhythmias. Further investigation of arrhythmia complications will be necessary before translating into human patients. As mentioned prior, electrical stimulation to enhance hCMP also requires a solution to stop causing arrhythmias [66]. The hypothesis that autologous cell transplantation may not elicit an immune response has been challenged by several recent studies, so caution must be exercised. For a long time in clinical practice, inhibition of the immune response has been mainly due to the use of immunosuppressive drugs such as cyclosporine, dexamethasone, and FK-506. Due to the side effects induced by such immunosuppressive drugs, there is a trade-off between efficacy and toxicity: high doses of immunosuppression cause complications associated with toxicity, and low doses of immunosuppression cause allotransplant rejection. However, many methods have been tested to control the immune response; immune tolerance may be induced through host regulation by activation/adoption of regulatory T cells. Moreover, modern genome editing tools can reduce immunogenicity by combining the inactivation of the MHC gene with the overexpression of CD47 [67]. CD47 is an omnipresent membrane protein that directly controls T-cell immunity by disrupting genes in HLA. These genetically modified hiPSCs can be used to generate stockpiles of “ready-made” heart cells, as well as hCMPs for emergency administration, without the need for concomitant immunosuppressive therapy.

## 5. Summary—Current Challenges and Future Prospects

The potential benefits of hCMP for the treatment of MI are readily observable in preclinical studies, and at least one small trial in patients is currently underway [68]. Traditional methods for producing hCMP include suspending cells in scaffolds of biocompatible materials or growing 2D sheets in culture and stacking them to form multilayer structures. More advanced techniques such as micropatterning and CAD-guided 3D bioprinting have allowed researchers to control the architecture of hCMP at a resolution that matches the scale of the interaction between individual cells or between cells and ECM. However, most studies have been conducted using a relatively small amount of hCMP that is not suitable for administration to patients. The size and (especially) thickness of hCMP are often limited by diffusion limitations of oxygen, nutrients, and other bioactive molecules, and CM must be within a distance of 100–200 μm from capillaries. Other challenges include the large amounts of CM required, the need for long-term graft retention, and the necessity for electromechanical coupling between the hCMP and the host tissue. The scientific community must address these issues, as well as the occurrence of arrhythmias, using large animal models before moving into human clinical trials.

## Figures and Tables

**Figure 1 bioengineering-09-00696-f001:**
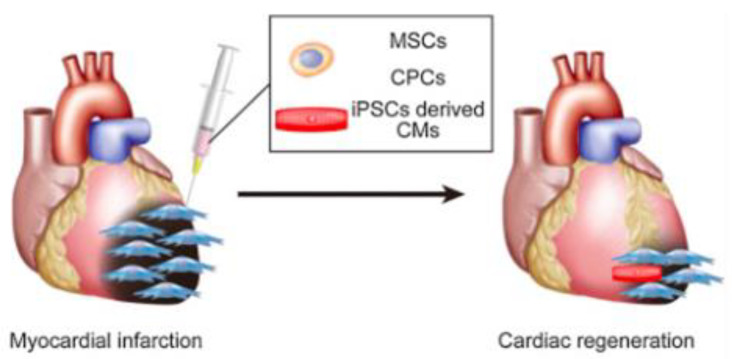
MSCs and CMs can be used to regenerate the myocardium [14].

**Figure 3 bioengineering-09-00696-f003:**
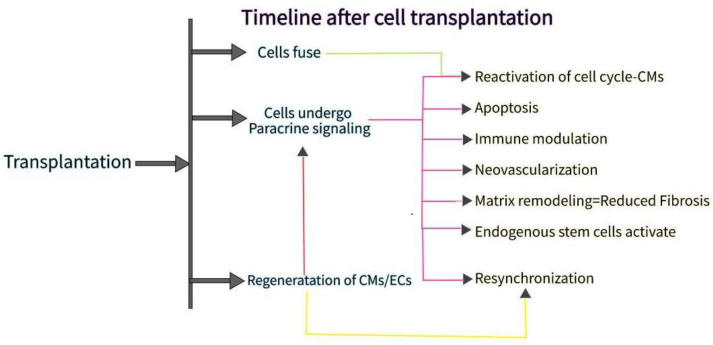
Transplanted cells often cause paracrine signaling, leading to apoptosis and reactivation of the cell cycle. These side effects often improve the regenerative ability of the heart.

## Data Availability

Not applicable.

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
