# Peer review of "Advances in Cardiac Tissue Engineering"

_bioengineering, 2022, doi:10.3390/bioengineering9110696_

Round 1

Reviewer 1 Report

This review summarized the advances in engineered CMPs. There are some specific comments on this manuscript 

1.     Reference should not be cited in Abstract

2.     In the last sentence, the second paragraph, Section 2, what is the [REF] meaning.

3.     The paragraph setting should be improved

4.     The Table1 should be improved, advantages, disadvantages, and applications can be added

5.     The title of 3.2 should be revised

6.     The clinical application of the CMPs should be added.

Author Response

Response to Reviewer #1

Thank you for the very useful advice. The responses to your comments

are described below.

Comment 1.1

1-1. Reference should not be cited in the Abstract

Response 1.1

We rewrote the abstract as follows: Tissue engineering has paved the way for the development of artificial human cardiac muscle patches (hCMPs) and cardiac tissue analogs, especially for treating Myocardial infarction(MI), often by increasing its regenerative abilities. Low engraftment rates, insufficient clinical application scalability, and the creation of a functional vascular system remain obstacles to hCMP implementation in clinical settings. This paper will address some of these challenges, present a broad variety of heart cell types and sources that can be applied to hCMP biomanufacturing, and describe some new innovative methods for engineering such treatments. It is also important to note the injection/transplantation of cells in cardiac tissue engineering,

Comment 1.2

1-2. In the last sentence, the second paragraph, Section 2, what is the [REF] meaning.

Response 1.2

We deleted all [REF], including the one in the second paragraph, and replaced them with references to the original article. It was a major mistake that we did not notice; thank you very much for your advice.

Comment 1.3

1-3. The paragraph setting should be improved

Response 1.3

Thank you for your advice; we have changed the paragraphs, images, and titles accordingly.

Comment 1.4

1-4. The Table1 should be improved, advantages, disadvantages, and applications can be added

Response 1.4

Following the reviewer’s comment, we added the advantages, disadvantages, and supporting cells to Table 1.

Comment 1.5

1.5. The title of 3.2 should be revised

Response 1.5

We changed the title of the paragraph to3D Printing; spheroids, contractile forces, and tubular EHTs”

Comment 1.6

1.6. The clinical application of the CMPs should be added.

Response 1.6

There is only 1 small trial in Osaka Univerity with the clinical application of hCMPs; however, we did add a reference to the original sentence explaining the trial.

Reviewer 2 Report

This review on the advances of tissue engineering describes novel findings in the fabrication of structures resembling the human heart. To my view, this manuscript has only limited news value. Also, the manuscript is incomplete as it lacks many references, one figure (figure 2), so it not written and edited very carefully. I have some major concerns and a few minor remarks.

Major remarks

1) The authors have insufficiently referred to important studies. Sometimes [ref] is used without really referring to a study. I have given some examples in the minor comments, but this is a general and important point.

2) Important advances, such as electromechanical stimulation to enhance cardiac tissue engineering have not been discussed.

Minor remarks

3) In the introduction when discussing the direct delivery of cells to the injured heart (cell therapy), please refer to some recent important studies, such as Liu - Nature Biotech - 2018

4) Important studies in engineered heart tissue published in high impact journals are not discussed:   Breckwoldt - Nat-Prot - 2017 ; Zhao - Cell - 2019 ; Ruan - Circ - 2016 

5) No references in abstract.

6) Incomplete referring: Some statements where references are needed (this list is not complete): 

line 52, 54, 57, 83, 89, 96, 99, 107, 108, 156, 162, 172, 185 etc.

7) Table 1 does not include the supporting cells.

8) figure 1b does not add much new information or insight, remove please. Also rethink the order of this, since you start introduction with use of cell therapy and then move on to tissue engineering. why not show fig 1d separately?

9) figure 2 is missing

10) figure 3 in unclear

Author Response

Response to Reviewer #2

Thank you for the very helpful advice. The responses to your comments are described

below.

Comment 2.1

2.1. Major remark: The authors have insufficiently referred to important studies. Sometimes [ref] is used without really referring to a study. I have given some examples in the minor comments, but this is a general and important point.

Response 2.1

We deleted all [REF]s and replaced them with references to the original article.

Comment 2.2

2.2. Major Remark: Important advances, such as electromechanical stimulation to enhance cardiac tissue engineering have not been discussed.

Response 2.2

We researched and added the discussion about electromechanical stimulation under 2.3, for electromechanical stimulations have the ability to grow CMs. We added this paragraph-

It is also worth noting how electrical stimulation or excitation-contraction coupling can affect the amount and duration of action potentials in CMs, thus increasing the number of synchronous contractions. Electrical stimulation can trigger several transcription factors such as NRF-1, GATA4, NFAT3, and cytochrome c, leading to increased growth and maturation in CMs. However, electrical stimulation could lead to arrhythmias due to the changes in electrical coupling. Moreover, most biomaterials used in 3D constructs are still not conductive, requiring new materials such as graphene to be added.

Comment 2.3

2.3. Minor remark: in the introduction when discussing the direct delivery of cells to the injured heart (cell therapy), please refer to some recent important studies, such as Liu - Nature Biotech - 2018. Important studies in engineered heart tissue published in high impact journals are not discussed:   Breckwoldt - Nat-Prot - 2017 ; Zhao - Cell - 2019 ; Ruan - Circ - 2016

Response 2.3

We read these articles and referenced said articles in the respected sentences and agreed that is was a good addition to our references.

Comment 2.4

2.4. Minor Remark: No references in the abstract

Response 2.4

We rewrote the abstract as follows: Tissue engineering has paved the way for the development of artificial human cardiac muscle patches (hCMPs) and cardiac tissue analogs, especially for treating Myocardial infarction(MI), often by increasing its regenerative abilities. Low engraftment rates, insufficient clinical application scalability, and the creation of a functional vascular system remain obstacles to hCMP implementation in clinical settings. This paper will address some of these challenges, present a broad variety of heart cell types and sources that can be applied to hCMP biomanufacturing, and describe some new innovative methods for engineering such treatments. It is also important to note the injection/transplantation of cells in cardiac tissue engineering.

Comment 2.5

2.5. Minor Remark: Incomplete referring: Some statements where references are needed (this list is not complete):

line 52, 54, 57, 83, 89, 96, 99, 107, 108, 156, 162, 172, 185 etc.

Response 2.5

We double-checked and inserted references in said areas, as well as additional lines that require references. We missed a few of them during the editing process, and could not thank you for your help in finding them for us.

Comment 2.4

2.4. Minor Remark: Table 1 does not include the supporting cells.

Response 2.4

We followed your advice and added the supporting cells in Table 1.

Comment 2.5

2.5. Minor Remark: figure 1b does not add much new information or insight, remove please. Also rethink the order of this, since you start introduction with use of cell therapy and then move on to tissue engineering. why not show fig 1d separately?

Response 2.5

We took this advice and deleted the previous Figure 1. b, as well as broke apart Figure 1.d and placed it above Figure 1. a and c.

Comment 2.6

2.6. Minor Remark: Figure 2 is missing

Response 2.6

We resolved this issue by making Figure 3 into Figure 2.

Comment 2.7

2.7. Minor Remark: figure 3 in unclear

Response 2.7

We agreed that the unparallel verbal usage with parts of speech made it difficult to understand, so we revised all the words into verb form and deleted a few lines. 

Reviewer 3 Report

well done to review the new development of cardiac tissue engineering.

Author Response

Dr. Daphne Hu

Assistant Editor

MDPI

Oct 30, 2022

Dear Dr. Hu,

Thank you for your e-mail dated Oct 18 2022 regarding our manuscript. We sincerely thank you for giving us the opportunity to revise our manuscript and for the critical comments that you and the reviewers provided.

We wish to resubmit our revised manuscript, “ Advances in Cardiac Tissue Engineering”, for publication in MDPI.

We revised our manuscript after careful consideration of the comments raised by the

reviewers. The changes we made in the revised manuscript are listed in the attached

point-by-point responses to the reviewers’ comments. All changes made in response to

the comments are shown in red in the revised document.

We believe that these changes have addressed all of the critiques raised by the referees

and that the revisions have improved our manuscript. We hope that the revised

manuscript will meet with your approval and can be accepted for publication in MDPI

Sincerely yours,

Kitsuka Takahiro, MD, Ph.D.

Responses to Editorial requests and Reviewers’ comments

We greatly appreciate the editorial requests and suggestions. We believe that the

manuscript has been significantly improved by these revisions. Below are our point-by-point responses to the editorial requests and reviewers’ comments. 

Response to Reviewer #1

Thank you for the very useful advice. The responses to your comments

are described below.

Comment 1.1

1-1. Reference should not be cited in the Abstract

Response 1.1

We rewrote the abstract as follows: Tissue engineering has paved the way for the development of artificial human cardiac muscle patches (hCMPs) and cardiac tissue analogs, especially for treating Myocardial infarction(MI), often by increasing its regenerative abilities. Low engraftment rates, insufficient clinical application scalability, and the creation of a functional vascular system remain obstacles to hCMP implementation in clinical settings. This paper will address some of these challenges, present a broad variety of heart cell types and sources that can be applied to hCMP biomanufacturing, and describe some new innovative methods for engineering such treatments. It is also important to note the injection/transplantation of cells in cardiac tissue engineering,

Comment 1.2

1-2. In the last sentence, the second paragraph, Section 2, what is the [REF] meaning.

Response 1.2

We deleted all [REF], including the one in the second paragraph, and replaced them with references to the original article. It was a major mistake that we did not notice; thank you very much for your advice.

Comment 1.3

1-3. The paragraph setting should be improved

Response 1.3

Thank you for your advice; we have changed the paragraphs, images, and titles accordingly.

Comment 1.4

1-4. The Table1 should be improved, advantages, disadvantages, and applications can be added

Response 1.4

Following the reviewer’s comment, we added the advantages, disadvantages, and supporting cells to Table 1.

Comment 1.5

1.5. The title of 3.2 should be revised

Response 1.5

We changed the title of the paragraph to3D Printing; spheroids, contractile forces, and tubular EHTs”

Comment 1.6

1.6. The clinical application of the CMPs should be added.

Response 1.6

There is only 1 small trial in Osaka Univerity with the clinical application of hCMPs; however, we did add a reference to the original sentence explaining the trial.

Response to Reviewer #2

Thank you for the very helpful advice. The responses to your comments are described

below.

Comment 2.1

2.1. Major remark: The authors have insufficiently referred to important studies. Sometimes [ref] is used without really referring to a study. I have given some examples in the minor comments, but this is a general and important point.

Response 2.1

We deleted all [REF]s and replaced them with references to the original article.

Comment 2.2

2.2. Major Remark: Important advances, such as electromechanical stimulation to enhance cardiac tissue engineering have not been discussed.

Response 2.2

We researched and added the discussion about electromechanical stimulation under 2.3, for electromechanical stimulations have the ability to grow CMs. We added this paragraph-

It is also worth noting how electrical stimulation or excitation-contraction coupling can affect the amount and duration of action potentials in CMs, thus increasing the number of synchronous contractions. Electrical stimulation can trigger several transcription factors such as NRF-1, GATA4, NFAT3, and cytochrome c, leading to increased growth and maturation in CMs. However, electrical stimulation could lead to arrhythmias due to the changes in electrical coupling. Moreover, most biomaterials used in 3D constructs are still not conductive, requiring new materials such as graphene to be added.

Comment 2.3

2.3. Minor remark: in the introduction when discussing the direct delivery of cells to the injured heart (cell therapy), please refer to some recent important studies, such as Liu - Nature Biotech - 2018. Important studies in engineered heart tissue published in high impact journals are not discussed:   Breckwoldt - Nat-Prot - 2017 ; Zhao - Cell - 2019 ; Ruan - Circ - 2016

Response 2.3

We read these articles and referenced said articles in the respected sentences and agreed that is was a good addition to our references.

Comment 2.4

2.4. Minor Remark: No references in the abstract

Response 2.4

We rewrote the abstract as follows: Tissue engineering has paved the way for the development of artificial human cardiac muscle patches (hCMPs) and cardiac tissue analogs, especially for treating Myocardial infarction(MI), often by increasing its regenerative abilities. Low engraftment rates, insufficient clinical application scalability, and the creation of a functional vascular system remain obstacles to hCMP implementation in clinical settings. This paper will address some of these challenges, present a broad variety of heart cell types and sources that can be applied to hCMP biomanufacturing, and describe some new innovative methods for engineering such treatments. It is also important to note the injection/transplantation of cells in cardiac tissue engineering.

Comment 2.5

2.5. Minor Remark: Incomplete referring: Some statements where references are needed (this list is not complete):

line 52, 54, 57, 83, 89, 96, 99, 107, 108, 156, 162, 172, 185 etc.

Response 2.5

We double-checked and inserted references in said areas, as well as additional lines that require references. We missed a few of them during the editing process, and could not thank you for your help in finding them for us.

Comment 2.4

2.4. Minor Remark: Table 1 does not include the supporting cells.

Response 2.4

We followed your advice and added the supporting cells in Table 1.

Comment 2.5

2.5. Minor Remark: figure 1b does not add much new information or insight, remove please. Also rethink the order of this, since you start introduction with use of cell therapy and then move on to tissue engineering. why not show fig 1d separately?

Response 2.5

We took this advice and deleted the previous Figure 1. b, as well as broke apart Figure 1.d and placed it above Figure 1. a and c.

Comment 2.6

2.6. Minor Remark: Figure 2 is missing

Response 2.6

We resolved this issue by making Figure 3 into Figure 2.

Comment 2.7

2.7. Minor Remark: figure 3 in unclear

Response 2.7

We agreed that the unparallel verbal usage with parts of speech made it difficult to understand, so we revised all the words into verb form and deleted a few lines. 

Response to Reviewer #3

Thank you for the very useful advice. The responses to your comments

are described below.

Comment 3

  1. Well done to review the new development of cardiac tissue engineering.

Reponse 3

We greatly appreciate your thoughts on our manuscript.

Round 2

Reviewer 1 Report

This manuscript is appropriate for publication

Author Response

We greatly appreciate your advice for this paper; thank you very much for the time and effort put in to revising our manuscript.

Reviewer 2 Report

The revised manuscript is still incomplete as it still lacks references it adviced in my first review. In my view, some important aspects of engineered heart tissue are completely neglected.  Important studies I previously advised are still not discussed.

 The authors failed to address comment 3 from my first review:

3) In the introduction when discussing the direct delivery of cells to the injured heart (cell therapy), please refer to some recent important studies, such as Liu - Nature Biotech - 2018

The authors failed to address comment 4 from my first review:

4) Important studies in engineered heart tissue published in high impact journals are not discussed:   Breckwoldt - Nat-Prot - 2017 ; Zhao - Cell - 2019 ; Ruan - Circ - 2016

The paper from breckwoldt et al is now referred too, but very poorly discussed. I do not understand why these high impact, and important advances in cardiac tissue engineering are not part of this review.

 Specific comments:

1. Introduction:

“Thus, surgery becomes the only treatment possible for none of the developed vaccines and oral treatments have been used clinically.”

The authors neglect the pharmacological treatment of heart failure. Also it is unclear what is meant by the developed vaccines and oral treatments.

2. cell types …

“In the heart, the newborn ratio for the number and volume ratio of CMs to non-CMs is 7:3. As the heart grows to adult size. However, the volume ratio does not change, the ratio for the number of CMs to non-CMs becomes 3:7 with the non-CMs increasing in proportion.”

What is the reference for this?

2.3 cardiomyocytes

“Moreover, most biomaterials used in 3D constructs are still not conductive, requiring new materials such as graphene to be added (24-26).”

I do not understand why the biomaterial needs to be conductive?

Figure 2

Legend is insufficiently clear

Author Response

Responses to Editorial requests and Reviewers’ comments
We greatly appreciate the editorial requests and suggestions. We believe that the 
manuscript has been significantly improved by these revisions. Below are our point-bypoint responses to the editorial requests and reviewers’ comments. Text in blue in this 
letter indicates the editorial requests and reviewers’ comments.
Response to Reviewer #1
Thank you for the very useful advice. The responses to your comments are described 
below.
Comment 1.1
1-1. In the introduction when discussing the direct delivery of cells to the injured heart 
(cell therapy), please refer to some recent important studies, such as Liu - Nature Biotech 
- 2018
Response 1.1
We added a few sentences describing Liu’s study in the introduction, and worked on 
smoothing out the transition to the next sentence. 
Comment 1.2
1-2. Important studies in engineered heart tissue published in high impact journals are 
not discussed: Breckwoldt - Nat-Prot - 2017 ; Zhao - Cell - 2019 ; Ruan - Circ - 2016
The paper from breckwoldt et al is now referred too, but very poorly discussed. I do not 
understand why these high impact, and important advances in cardiac tissue engineering 
are not part of this review.
Response 1.2
We agree that the Breckwoldt study was misplaced, and could have been discussed further. 
We referenced all of those journals, but also went further and included their studies in the 
review paper, to provide further evidence. 
Comment 1.3
1-3. “Thus, surgery becomes the only treatment possible for none of the developed 
vaccines and oral treatments have been used clinically.”
The authors neglect the pharmacological treatment of heart failure. Also it is unclear 
what is meant by the developed vaccines and oral treatments.
Response 1.3
We changed the wording to make it clear that heart transplantation was the only longterm and drastic improvement for myocardial infarction. 
Comment 1.4
1-4. “In the heart, the newborn ratio for the number and volume ratio of CMs to nonCMs is 7:3. As the heart grows to adult size. However, the volume ratio does not change, 
the ratio for the number of CMs to non-CMs becomes 3:7 with the non-CMs increasing 
in proportion.”
What is the reference for this?
Response 1.4
We successfully found the source for this information.
Comment 1.5
1-5. “Moreover, most biomaterials used in 3D constructs are still not conductive, 
requiring new materials such as graphene to be added (24-26).”
I do not understand why the biomaterial needs to be conductive?
Response 1.5
The biomaterial needs to be conductive in order to allow for electrical stimulation, but we 
changed the wording of the sentence to make it clearer to the reader. 
Comment 1.6
1-6. Legend is insufficiently clear
Response 1.6
We changed the wording of the legend, and also noticed that part of figure 2.a was cut 
out, so we replaced it with a new image. 
